# Elliptical Perturbations for Differential Privacy

**Matthew Reimherr** *
Department of Statistics
Pennsylvania State University
University Park, PA 16802
mreimherr@psu.edu

**Jordan Awan** †
Department of Statistics
Pennsylvania State University
University Park, PA 16802
awan@psu.edu

## Abstract

We study elliptical distributions in locally convex vector spaces, and determine conditions when they can or cannot be used to satisfy differential privacy (DP). A requisite condition for a sanitized statistical summary to satisfy DP is that the corresponding privacy mechanism must induce equivalent probability measures for all possible input databases. We show that elliptical distributions with the same dispersion operator, $C$, are equivalent if the difference of their means lies in the Cameron-Martin space of $C$. In the case of releasing finite-dimensional summaries using elliptical perturbations, we show that the privacy parameter $\epsilon$ can be computed in terms of a one-dimensional maximization problem. We apply this result to consider multivariate Laplace, $t$, Gaussian, and $K$-norm noise. Surprisingly, we show that the multivariate Laplace noise does not achieve $\epsilon$-DP in any dimension greater than one. Finally, we show that when the dimension of the space is infinite, no elliptical distribution can be used to give $\epsilon$-DP; only $(\epsilon, \delta)$-DP is possible.

## 1 Introduction

Infinite dimensional objects and parameters arise commonly in nonparametric statistics, shape analysis, and functional data analysis. Several recent works have made strides towards extending tools for differential privacy (DP) to handle such settings. Some of the first results in this area were given in Hall et al. (2013), with a particular emphasis on Gaussian perturbations and point-wise releases of statistical summaries represented as univariate functions. This work was extended to more general Banach and Hilbert space based summaries by Mirshani et al. (2017), which included protections for public releases based on path level summaries, nonlinear transformations of functional summaries, and full function releases as well. However, Gaussian perturbations are not always satisfactory since they cannot be used to achieve pure DP ($\epsilon$-DP), which requires heavier tailed distributions. Rather, for pure DP, the most popular distribution is the Laplace mechanism, whose tails are "just right" for achieving DP in finite dimensional summaries (Dwork et al., 2006).

When one moves from univariate to multivariate settings, generalizing the Laplace mechanism is not as simple as generalizing the Gaussian. Often, when the Laplace mechanism is used in multivariate settings, iid Laplace random variables are used. However, this approach fails to capture the multivariate dependence structure of the data or parameter of interest. Furthermore, in infinite dimensional settings, adding iid noise is usually not an option if one wishes to remain in a particular function space. To address these issues, we study the use of elliptical distributions to satisfy DP, which allow for a dispersion operator and are closely related to Gaussian distributions. Elliptical

processes offer a nice option for designing DP mechanisms for multivariate and infinite-dimensional data as they allow for the customization of both the tail behavior and dependence structure, which can be tailored to the problem at hand. Recently Bun and Steinke (2019) explored several alternative univariate distributions for achieving privacy such as Cauchy, Student's T, Laplace Log-Normal, Uniform Log-Normal, and Arsinh-Normal, which can be extended to elliptical distributions.

We are interested in releasing a sanitized version of a statistic $T : \mathcal{D} \to \mathbb{X}$, where $\mathcal{D}$ is a metric space, representing the space of possible input databases, $D$, and $\mathbb{X}$ is a locally convex vector space. To achieve differential privacy, we will release $\widetilde{T} = T(D) + \sigma X$, where $\sigma$ is a positive scalar, and $X$ is a random element of $\mathbb{X}$. In particular, we consider $X$ which are drawn from elliptical distributions, of which the multivariate Laplace and Gaussian distributions are special cases. Most linear spaces are locally convex vector spaces, including all Hilbert Spaces, Banach Spaces, Frechet spaces, and product spaces of normed vector spaces, meaning that our results will hold quite broadly.

We consider the setting where the statistical summary and privacy mechanism are *truly infinite dimensional*, meaning that the problem cannot be embedded into a finite dimensional subspace where multivariate privacy tools can be used. There are both interesting mathematical and practical motivations for this perspective. First, our setting can be viewed as a limit of multivariate problems; if one has privacy over the full infinite dimensional space, then this ensures that the noise is well behaved when releasing multivariate summaries, regardless of how many are released. Second, one does not need to ensure that every database uses the same finite dimensional subspace, allowing practitioners to use whatever methods and summaries they prefer. And third, our setting is very convenient when addressing multiple queries. In particular, one does not need to spend a fraction of the privacy budget for every query. Instead, the amount spent for each subsequent query decreases dramatically, eventually leveling out to a maximum $\epsilon$ or $(\epsilon, \delta)$. To accomplish this, one does not need to "store" the infinite dimensional noise, instead, we can generate as much of the noise as is needed for a particular query while conditioning on any noise values generated for prior queries.

We also provide a surprising result showing that $\epsilon$-DP can only be achieved for a finite number of summaries or point-wise evaluations; in infinite dimensions no elliptical perturbation is capable of achieving $\epsilon$-DP over the full function space, one can only achieve $(\epsilon, \delta)$-DP. This is in stark contrast with what is known from the univariate or multivariate literature on DP.

While elliptical distributions are being used more frequently in statistics and machine learning (e.g. Schmidt, 2003; Frahm et al., 2003; Soloveychik and Wiesel, 2014; Couillet et al., 2016; Sun et al., 2016; Goes et al., 2017; Ollila and Raninen, 2019), some fundamental questions regarding elliptical distributions in function spaces remain underdeveloped. For data privacy, the question of equivalence/orthogonality of elliptical measures is particularly important. In terms of data privacy, if a perturbation in a dataset produces a private summary that is orthogonal (in a probabilistic sense) to the old one, then the summaries cannot be differentialy private since they can be distinguished with probability one. We show that several conditions for making this determination transfer nicely from the Gaussian setting, but not all. While conditions on the location function remain the same, conditions on the dispersion function change. Furthermore, that all elliptical measures are equivalent or orthogonal need no longer hold without additional assumptions. Regardless, for the purposes of privacy, determining equivalence/orthogonality based on the location is the primary requirement.

**Related Work:** Our general approach of adding noise from a data-independent distribution to a summary statistic is one of the simplest and most common methods of achieving DP. This approach was first developed using the Laplace mechanism (Dwork et al., 2006), and has since been expanded to include a larger variety of distributions. Ghosh et al. (2009) showed that when the data is a count, then the optimal noise adding distribution is discrete Laplace. Geng and Viswanath (2016) extended this result to the continuous setting, developing the staircase distribution which is closely related to discrete Laplace. In the multivariate setting, the most common solution is to add iid Laplace noise to each coordinate (Dwork and Roth, 2014). However, Hardt and Talwar (2010) and Awan and Slavković (2019) demonstrate that capturing the covariance structure in the data, via the $K$-norm mechanism can substantially reduce the amount of noise required.

After adding noise to summary statistics, researchers have shown that many complex statistical and machine learning tasks can be produced by post-processing, such as linear regression (Zhang et al., 2012), maximum likelihood estimation (Karwa and Slavković, 2016), hypothesis testing (Vu and Slavković, 2009; Gaboardi et al., 2016; Awan and Slavković, 2018), posterior inference (Williams and Mcsherry, 2010; Karwa et al., 2016), or general asymptotic analysis (Wang et al., 2018).

To date, the only additive mechanism in infinite-dimensions is the Gaussian mechanism, developed by Hall et al. (2013) and Mirshani et al. (2017). However, there has been other work on developing privacy tools for these spaces. Awan et al. (2019) show that the exponential mechanism (McSherry and Talwar, 2007) can be used in arbitrary Hilbert spaces, by integrating with respect to a fixed probability measure such as a Gaussian process. An alternative approach proposed by Alda and Rubinstein (2017) uses Bernstein polynomial approximations to release private functions. Recently, Smith et al. (2018) utilized the techniques of Hall et al. (2013) to develop private Gaussian process regression. Similar to the pufferfish approach (Kifer and Machanavajjhala, 2014), they assume that the predictors are public, and use the known covariance structure to tailor the noise distribution.

**Organization:** In Section 2, we review the necessary background on locally convex vector spaces, elliptical distributions, and differential privacy. In Section 3, we study the equivalence and orthogonality of elliptical measures, and give a condition that ensures that two elliptical measures are equivalent. In Section 4, we investigate using elliptical perturbations to achieve DP. First, we consider the finite dimensional case in Section 4.1, and in Theorem 3 we give a condition for elliptical perturbations to satisfy $\epsilon$-DP as well as a method of computing $\epsilon$. In Section 4.2 we show that if the dimension of the space is infinite, then no elliptical perturbation can satisfy $\epsilon$-DP. In fact, we show that every elliptical distribution can only achieve $(\epsilon, \delta)$-DP for a positive $\delta$. We give short proof sketches throughout the document, with detailed proofs left to the Supplementary Material.

## 2    Elliptical Distributions

Elliptical distributions, whether over $\mathbb{R}^d$ or a more general vector space can be defined in a variety of equivalent ways. Intuitively, an elliptical distribution is one in which its density contours form hyperellipses. However, this presupposes that the measure is absolutely continuous with respect to Lebesgue measure. Thus, it is often useful in multivariate settings to use alternative definitions that are more easy to generalize, but which are equivalent to the shape of the density contours when they exist. This is not unique to elliptical measures, such alternative definitions are often useful when working with infinite dimensional objects (e.g. Bosq, 2000). Throughout, we focus our attention on an arbitrary, real, locally convex vector space (from here on LCS), $\mathbb{X}$, but we will restrict ourselves to simpler spaces (e.g. Banach, Hilbert, or Euclidean spaces) as needed or for illustration. For ease of reference, recall the following concepts from functional analysis (see Rudin (1991) for an introduction).

- A set, $\mathbb{X}$, is called a *vector space* if it is closed under addition and scalar multiplication (and those operations are well defined).

- A vector space, $\mathbb{X}$, is called a *topological vector space*, if it is equipped with a topology under which addition and scalar multiplication are continuous.

- A topological vector space $\mathbb{X}$ is called *locally convex* if its topology is generated by a separated family of semi-norms, $\{p_\alpha : \alpha \in \mathcal{I}\}$, where $\mathcal{I}$ is an arbitrary index set and separated means that for all nonzero $x \in \mathbb{X}$ there exists $\alpha \in \mathcal{I}$ such that $p_\alpha(x) \neq 0$. A base for the topology is given by sets of the form $A_{\alpha, \epsilon} = \{x \in \mathbb{X} : p_\alpha(x) < \epsilon\}$.

- The topological dual, $\mathbb{X}^*$, is the collection of all continuous linear functionals on $\mathbb{X}$.

The assumption that the seminorms are separated is not always included in the definition, but is equivalent to assuming that the space is Hausdorff. Recall that a topology defines the open sets, a collection of subsets that is closed under uncountable unions, finite intersections, and contains both $\mathbb{X}$ and $\emptyset$. We use this level of generality to include as many settings as possible into our framework. In particular, all finite dimensional Euclidean spaces, normed vector spaces, Hilbert Spaces, Banach Spaces, and Frechet spaces are types of LCS. In addition, uncountable product spaces of normed spaces, which are often used in the mathematical foundations of stochastic processes, are LCS as well (when equipped with the product topology). To find practical examples of spaces that are not locally convex spaces, one either has to consider nonlinear spaces, such as manifolds, or equip a space with an "odd" metric (such as $L^p$ for $p < 1$).

**Example 1** (LCS Examples)**.** The definition of LCS in terms of seminorms is perhaps unintuitive at first, but can be motivated by product spaces such as $\mathbb{R}^{[0,1]} = \{f : [0,1] \to \mathbb{R}\}$. The space $\mathbb{R}^{[0,1]}$ is in a sense "too large" to accommodate a norm, but it is easy to define a family of semi-norms that measure the magnitude of the function coordinate-wise. Here $\alpha \in \mathcal{I} := [0,1]$ and we define

$p_\alpha(f) = |f(\alpha)|$. Note that for any particular $\alpha$, $p_\alpha$ is not a norm, since $p_\alpha(f-g) = 0$ does not imply that $f = g$. However, the entire collection of semi-norms *separates points* since $p_\alpha(f-g) = 0$ for all $\alpha \in [0,1]$ implies that $f = g$.

It is easy to see that any normed space fits the definition of LCS. For $C[0,1]$, we set $\mathcal{I} = \{1\}$ and define $p_1(f) = \sup_{t\in[0,1]} |f(t)|$, and for $L^2[0,1]$ we set $\mathcal{I} = \{1\}$ and define $p_1(f) = \int f^2(t)dt$.

When working with a LCS one commonly uses one of two $\sigma$-algebras. The Borel $\sigma$-algebra, $\mathcal{B}$, is the smallest $\sigma$-algebra that contains the open sets. The cylinder $\sigma$-algebra, $\mathcal{C}$, is the smallest $\sigma$-algebra that makes all continuous linear functionals measurable. In general we have $\mathcal{C} \subseteq \mathcal{B}$, but these two $\sigma$-algebras are not equal unless the space has additional structure, e.g. separable Banach spaces. This creates complications in infinite dimensional settings. For example, the technical theory for stochastic processes often starts with product spaces such as $\mathbb{R}^{[0,1]}$. There, the two sigma algebras are not the same, which is an issue for privacy as one desires privacy over $\mathcal{B}$, not just $\mathcal{C}$. This is because only the events in the chosen $\sigma$-algebra are protected, and a larger $\sigma$-algebra offers a stronger privacy guarantee. More importantly, $\mathcal{C}$ does not contain most sets of interest, including continuous functions, linear functions, polynomials, constant functions, etc. (Billingsley, 1979, Problem 36.6). To overcome this challenge, Mirshani et al. (2017) used Cameron-Martin theory, to obtain DP over all of $\mathcal{B}$ through careful use of densities in infinite dimensional spaces. This theory is built upon Gaussian processes; however, we will show that several of their key results, especially those needed for privacy, extend directly to elliptical distributions. Throughout this paper, we assume $\mathbb{X}$ is equipped with its Borel $\sigma$-algebra when discussing measures, measurability, and DP.

Often it is convenient to define probability measures over abstract spaces in terms of their characteristic functionals (i.e. Fourier transforms), which uniquely determine measures in any LCS.

**Definition 1** (Fang, 2017). A measure, $P$, over a locally convex space $\mathbb{X}$ is called elliptical if and only if its characteristic functional, $\widetilde{P} : \mathbb{X}^* \to \mathbb{C}$, has the form

$$\widetilde{P}(g) = \int_{\mathbb{X}} \exp\{ig(x)\}\, dP(x) = e^{ig(\mu)}\phi_0(C(g,g)),$$

where $\mu \in \mathbb{X}$, $C$ is a symmetric, positive definite, continuous bilinear form on $\mathbb{X}^* \times \mathbb{X}^*$, and $\phi_0$ is a positive definite function over $\mathbb{R}$, which is continuous at 0 and satisfies $\phi_0(0) = 1$.

Definition 1 implies that the distribution of $P$ is uniquely determined by knowing $\mu$, $C$, and $\phi_0$. The object $\mu$ denotes the center of the distribution; we will say a distribution is centered if $\mu = 0$. The object $C$ is often called the covariance or dispersion operator. In general, $C$ can either be identified as an operator or bilinear form (in fact a $(0,2)$ tensor). We avoid introducing extra notation we will let $C(g)$ denote the operator version and $C(f,g)$ the bilinear form.

We begin by presenting a second characterization of elliptical measures. This is a well known result, but we are unaware of a reference for this level of generality. We cite Fang (2017), which covers the multivariate case, but the proof is the same for general LCS.

**Theorem 1** (Fang, 2017). *Let $X \in \mathbb{X}$ be an elliptically distributed random variable. Then there exists a mean zero Gaussian process $Z \in \mathbb{X}$ with covariance operator $C$, an element $\mu \in \mathbb{X}$, and a strictly positive random variable $V \in \mathbb{R}^+$, that is independent of $Z$, and satisfies $X \overset{\mathcal{L}}{=} \mu + VZ$.*

This result is often phrased as "every elliptical distribution is a scalar mixture of Gaussian processes." While it is, of course, a fascinating result in its own right, it also provides a simple method of generating and simulating from arbitrary elliptical distributions.

Due to this corollary, we will index every elliptical measure using $\mu$, $C$, and the mixing distribution of $V$, which we will denote as $\psi$, and use the notation $\mathcal{E}(\mu, C, \psi)$. Equivalently, we could index using the $\phi_0$ from Definition 1, but our results in Section 3 are easier to present in terms of $\psi$.

We conclude by stating a general definition of DP, which makes sense over any measurable space, though we state it here for LCS. The concept of differential privacy was first introduced in Dwork et al. (2006) and Dwork (2006). Over time researchers have worked to make the definition more precise and flexible, such as Wasserman and Zhou (2010) who state it in terms of conditional distributions. For a general, axiomatic treatment of formal privacy, see Kifer et al. (2012).

**Definition 2.** Let $(\mathcal{D}, d)$ be a metric space and $\{P_D : D \in \mathcal{D}\}$ be a family of probability measures over a locally convex topological vector space $\mathbb{X}$. We say the family achieves $(\epsilon, \delta)$-differential

privacy if for any $d(D, D') \leq 1$ and any measurable $A$, we have

$$P_D(A) \leq e^\epsilon P_{D'}(A) + \delta. \tag{1}$$

Intuitively, $\mathcal{D}$ represents the universe of possible input databases. One then refers to $\{P_D : D \in \mathcal{D}\}$ as the *privacy mechanism*. The most common setting when discussing DP is when $\mathcal{D}$ is a product space and the metric is the Hamming distance. However, the Hamming distance (which counts differences in coordinates) is insensitive to the magnitude of the difference between two inputs $D$ and $D'$, thus one may wish to consider alternatives and so we take the more general approach. As discussed earlier, while DP can be defined with any $\sigma$-algebra, we assume that $\mathbb{X}$ is equipped with the Borel $\sigma$-algebra as it offers more intuitive guarantees. We refer to $(\epsilon, \delta)$-DP as *approximate DP* when $\delta > 0$ and as *pure-DP* when $\delta = 0$. When using pure DP, we often just write $\epsilon$-DP.

Another way of viewing $\epsilon$-DP (that is, taking $\delta = 0$) is through the equivalence/orthogonality of probability measures. As was discussed in Awan et al. (2019), in an $\epsilon$-DP mechanism the individual measures that make up the mechanism are all equivalent in a probabilistic sense (meaning they agree on the zero sets). Conversely, if the measures are orthogonal then the mechanism cannot even be $(\epsilon, \delta)$-DP. This perspective was used in Mirshani et al. (2017) for the case of Gaussian mechanisms. However, the corresponding theory for elliptical distributions is less developed. In the next section we extend several fundamental results of Gaussian processes to elliptical distributions.

## 3  Equivalence and Orthogonality of Elliptical Measures

A classic result from probability theory is that any two Gaussian processes are either equivalent or orthogonal (that is, as probability measures they either agree on the zero sets or concentrate their mass on disjoint sets). Recall that by the Radon-Nikodym theorem, if two measures are equivalent then there exists a density of one with respect to the other (and vice versa). What we will now show is that this property, to a degree, extends to any elliptical family. Furthermore, we will show that the conditions for establishing this equivalence/orthogonality are nearly the same as for Gaussian processes. We begin with a fairly simple yet surprisingly useful technical lemma.

**Lemma 1.** *Let $(\Omega, \mathcal{F}, P)$ be a probability space. Let $X^1 : \Omega \to \mathbb{X}$ and $X^2 : \Omega \to \mathbb{X}$ denote two random elements of $\mathbb{X}$, and let $T^1 : \Omega \to \mathbb{R}$ and $T^2 : \Omega \to \mathbb{R}$ be two random variables. Let $P^i$ denote the probability measure over $\mathbb{X}$ induced by $X^i$ and let $Q^i$ denote the measure over $\mathbb{R}$ induced by $T^i$. Let $P_t^i$ denote the conditional measure of $X^i$ given $T^i = t$. If $Q^1$ and $Q^2$ are equivalent and $P_t^1$ and $P_t^2$ are equivalent for almost all $t$ (wrt $Q^1$) then so are $P^1$ and $P^2$.*

The proof of Lemma 1 is in the Supplementary Material. Implicit in Lemma 1 is that the conditional distributions exist. This is not an issue in our setting as the conditional distributions can be explicitly constructed for elliptical processes, however, for general processes and spaces one can encounter nontrivial technical problems. We refer the interested reader to Hoffmann-Jørgensen (1972), Bogachev (1998, THM A.3.11), and Kallenberg (2006, Chapter 6) for further discussion. Interestingly, the reverse statement is not true. That is, even if all of the conditional distributions are orthogonal, the unconditional measures need not be orthogonal. To see this, suppose that $T$ is 0 or 1 with equal probability. Now, assume that $P_0^1$ and $P_1^1$ are orthogonal and set $P_0^2 = P_1^1$ and $P_1^2 = P_0^1$. Clearly the conditional distributions are orthogonal, but not only are the unconditional measures equivalent, they are actually the same!

Regardless, our goal is more specific; we want to establish conditions under which $\mathcal{E}(\mu_1, C, \psi)$ and $\mathcal{E}(\mu_2, C, \psi)$ are orthogonal when they share the same $\psi$ and $C$. In terms of DP, $\mu_1$ and $\mu_2$ represent the private summary from two different databases. If $\psi$ is a point mass, then the two measures are Gaussian and the conditions are known. The question is, to what degree do such conditions extend to other mixtures? Theorem 2 shows that the same conditions for Gaussian processes (with the same covariance, but different means) apply to any elliptical family. Given Corollary 1, this may seem obvious, but Lemma 1 implies that the matter is surprisingly delicate. For example, two Gaussian processes with the same mean, but where one has a covariance equal to a scalar $c \neq 1$ multiple of the other, are actually orthogonal (in infinite dimensions). This need not hold for arbitrary elliptical families as the scalar can be absorbed by the mixing coefficient (and then apply Lemma 1).

Our first major result establishes a condition under which DP cannot be achieved, regardless of the magnitude of the noise. First, let us define a subspace of $\mathbb{X}$ using the bilinear form $C$ (more

detail can be found in Bogachev (1998); Mirshani et al. (2017)). In particular, $C$ induces an inner product $\langle \cdot, \cdot \rangle_{\mathcal{K}}$ on the dual space $\mathbb{X}^*$ given by $\langle f, g \rangle_{\mathcal{K}} := C(f, g) = \int f(x)g(x)dP(x)$, where $P$ is a Gaussian measure with mean zero and covariance $C$. Then, we can view $\mathbb{X}^*$ as a subspace of $L^2(\mathbb{X}, P)$, the space of $P$-square integrable functions from $\mathbb{X} \to \mathbb{R}$. By assumption, $\langle \cdot, \cdot \rangle_{\mathcal{K}}$ is a continuous, symmetric, and positive definite bilinear form and thus a valid inner product. However, $\mathbb{X}^*$ is not complete with respect to this inner product when $\mathbb{X}$ is infinite dimensional, so let $\mathcal{K}$ denote its completion. Finally, consider the subset $\mathbb{H} \subset \mathbb{X}$, such that for $h \in \mathbb{H}$ the operation $T_h : \mathcal{K} \to \mathbb{R}$ given by $T_h(g) := g(h)$ is continuous in the $\mathcal{K}$ topology. Then $\mathbb{H}$ is called the *Cameron-Martin space* of $C$ (or equivalently, of the mean zero Gaussian process with $C$ as its covariance). Intuitively, the functionals in $\mathcal{K}$ are much "rougher" than those in $\mathbb{X}^*$ and thus the elements of $\mathbb{H}$ are much more regular than general elements of $\mathbb{X}$ to counter balance this. In fact, $C$ also generates an operator from $\mathcal{K} \to \mathbb{H}$ denoted as $C(g) = \int xg(x)dP(x)$. Using this notation, an element $h \in \mathbb{X}$ lies in $\mathbb{H}$ exactly when it equals $h = C(g)$ for some $g \in \mathcal{K}$. The space $\mathbb{H}$ is also a Hilbert space (even though $\mathbb{X}$ need not be) equipped with the inner product $\langle h_1, h_2 \rangle_{\mathbb{H}} = \langle g_1, g_2 \rangle_{\mathcal{K}}$ where $h_i = C(g_i)$.

**Theorem 2.** *Let $P_1 \sim \mathcal{E}(\mu_1, C, \psi)$ and $P_2 \sim \mathcal{E}(\mu_2, C, \psi)$ be two elliptical measures over a locally convex topological vector space, $\mathbb{X}$. Then the two distributions are equivalent if $\mu_1 - \mu_2$ resides in the Cameron-Martin space of $C$ and orthogonal otherwise.*

*Proof Sketch.* For the first direction, if $\mu_1 - \mu_2$ resides in the Cameron-Martin space of $C$ then it resides in the Cameron-Martin space of $vC$ for $v > 0$ since they induce equivalent norms. From Bogachev (1998, Theorem 2.4.5 ), two Gaussian measures with the same covariance, $C$, are equivalent if the difference of their means resides in the Cameron-Martin space of $C$. Thus, conditioned on the mixture $V = v$, the measures are equivalent for all $v$. By Lemma 1, they are equivalent.

For the reverse direction we consider, without loss of generality, $X_1 \sim \mathcal{E}(0, C, \psi)$ versus $X_2 \sim \mathcal{E}(\mu, C, \psi)$ where $\mu$ is not in the Cameron-Martin space of $C$. To see that the two measures are orthogonal, it suffices to show that, for any fixed $\epsilon \in (0, 1)$ we can construct a measurable set $A$ such that $P(X_1 \in A) \geq 1 - \epsilon$ while $P(X_2 \in A) \leq \epsilon$. $\qquad \square$

To interpret Theorem 2 in the context of privacy, given a database $D \in \mathcal{D}$, recall that a private summary is drawn from the elliptical distribution $\mathcal{E}(\mu_D, C, \psi)$. Theorem 2 then says that the measures are orthogonal (and thus no amount of noise will produce a DP summary) unless all of the differences $\mu_D - \mu_{D'}$, for any $D, D' \in \mathcal{D}$ reside in the Cameron-Martin space of $C$.

## 4 Achieving DP with Elliptical Perturbations

Now that we have the necessary tools in place and we know when we cannot have DP, we will now construct a broad class of mechanisms that do achieve DP. Recall that the mechanisms will be of the form $\widetilde{T}_D = T_D + \sigma X$, where $T_D := T(D)$ is the nonprivate statistical summary, $X$ is a prespecified elliptical process and $\sigma > 0$ is a fixed scalar. The exact value of $\sigma$ will be set to achieve some desired level of privacy. Gaussian perturbations (i.e. taking $\phi$ as a point mass) will not achieve $\epsilon$-DP even in finite dimensions. As is known in the literature, Gaussian perturbations have tails that are too light, causing the probability inequality of DP to fail for sets in the tails. To fix this, it is common to use another distribution, often the Laplace distribution, whose tails appear to be just right for achieving DP. Interestingly, this trick does not carry over to infinite dimensional spaces. We will show that while some elliptical distributions can achieve $\epsilon$-DP for finite dimensional projections, none can achieve it over the entire infinite-dimensional space; they can only achieve $(\epsilon, \delta)$-DP with $\delta > 0$.

### 4.1 DP in Finite Dimensions

In this subsection, we give a criterion (Theorem 3) that establishes which elliptical distributions satisfy $\epsilon$-DP, when $\mathbb{X} = \mathbb{R}^d$. We also provide a related result (Corollary 1) for $\epsilon$-DP with $d$-dimensional projections of infinite dimensional summaries, which holds uniformly across the choice of projection, for a fixed $d$. Elliptical distributions that can achieve $\epsilon$-DP (with a fixed $d$) include $\ell_2$-mechanism (Chaudhuri and Monteleoni, 2009; Chaudhuri et al., 2011; Kifer et al., 2012; Song et al., 2013; Yu et al., 2014; Awan and Slavković, 2019), and the multivariate $t$ distribution. Interestingly, the multivariate Laplace distribution cannot achieve $\epsilon$-DP when $d \geq 2$.

Denote by $\Sigma = \{C(e_i, e_j)\}$ the positive definite matrix containing the evaluations of $C$ on the standard basis of $\mathbb{R}^d$. Then the density of $\widetilde{T}_D = T_D + \sigma X$ is proportional to $f(\sigma^{-2}(x - T_D)\Sigma^{-1}(x - T_D))$, where $f$ is a decreasing positive function depending only on the dimension $d$ and the elliptical family for $X$. The omitted constants depend on $\Sigma$, but not on $T_D$. The Cameron-Martin norm can be expressed as $\|g\|_{\mathcal{H}} = g^\top \Sigma^{-1} g$. In fact $\mathcal{H} = \mathbb{R}^d$, but equipped with a different norm.

**Theorem 3.** *Assume that $\mathbb{X} = \mathbb{R}^d$ and, without loss of generality, assume that that $\widetilde{T}_D$ has a density with respect to Lebesgue measure proportional to $f_{\widetilde{T}_D}(x) \propto f(\sigma^{-2}(x - T_D)^\top \Sigma^{-1}(x - T_D))$, where $f : [0, \infty) \to [0, \infty]$ is a decreasing positive function. Set*

$$\Delta = \sup_{D \sim D'} \|T_D - T_{D'}\|_{\mathcal{H}} = \sup_{D \sim D'} \|\Sigma^{-1/2}(T_D - T_{D'})\|_2.$$

*If $\Delta < \infty$, $f(0) < \infty$, and*

$$\limsup_{c \to \infty} \frac{f((c - \Delta)^2)}{f(c^2)} < \infty, \tag{2}$$

*then $\widetilde{T}_D$ satisfies $\epsilon$-DP, where $\exp(\epsilon) = \sup_{c \geq \sigma^{-1}\Delta} \frac{f((c - \sigma^{-2}\Delta)^2)}{f(c^2)} < \infty.$*

The proof of Theorem 3 is based on the ratio of the densities, and is in the Supplementary Materials. Next we apply Theorem 3 to several common distributions.

**Example 2** (Independent Laplace)**.** Independent Laplace random variables are a common tool for achieving $\epsilon$-DP. The density of this mechanism is proportional to $f(x) \propto \exp\left(-\sum_{i=1}^d |x_i - \mu_i|/\sigma_i\right)$. While it is easily proved that this mechanism can be used to satisfy $\epsilon$-DP, this distribution is not elliptical, since the density cannot be written as a function of $(x - \mu)^\top \Sigma^{-1}(x - \mu)$ for any $\mu$ and $\Sigma$.

A natural idea is to use the elliptical multivariate Laplace distribution to try to achieve $\epsilon$-DP for multi-dimensional outputs. Surprisingly, the following example shows that while the tail behavior of the multivariate Laplace is sufficient to satisfy (2), the multivariate Laplace distribution cannot be used to achieve $\epsilon$-DP when $d \geq 2$, since it has a pole (i.e. goes to infinity) at its center.

**Example 3** (Multivariate Laplace)**.** A $d$-dimensional random variable $X \sim \text{Laplace}(\mu, \Sigma)$ has density equal to

$$2(2\pi)^{-d/2}|\Sigma|^{-1/2}\left((x - \mu)^\top \Sigma^{-1}(x - \mu)/2\right)^{\nu/2} K_\nu(\sqrt{2(x - \mu)^\top \Sigma^{-1}(x - \mu)}),$$

where $\nu = \frac{2-d}{2}$ and $K_\nu$ is the modified Bessel function of the second kind. This density is proportional to $f((x - \mu)^\top \Sigma^{-1}(x - \mu))$, where $f(y) = (y/2)^{\nu/2} K_\nu(\sqrt{2y})$. The reason this distribution is called the multivariate Laplace distribution is that it is the only family of distributions such that every marginal distribution is also distributed as Laplace (iid Laplace does not have this property).

First, let's check whether (2) is finite. We use the fact that $K_\nu(z) = c\exp(-z)z^{-1/2}(1 + O(1/z))$ as $z \to \infty$, where $c$ is a constant (Abramowitz and Stegun, 1965, Chapter 9). Then

$$\lim_{c \to \infty} \frac{f((c - \Delta)^2)}{f(c^2)} = \lim_{c \to \infty} \frac{\left(\frac{c-\Delta}{2}\right)^\nu K_\nu(\sqrt{2}(c - \Delta))}{\left(\frac{c}{2}\right)^\nu K_\nu(\sqrt{2}c)} = \lim_{c \to \infty} \frac{\exp(-\sqrt{2}(c - \Delta))\sqrt{c}}{\exp(-\sqrt{2}c)\sqrt{c - \Delta}} = \exp(\sqrt{2}\Delta).$$

We see that the tails of the multivariate Laplace distribution are heavy enough to satisfy $\epsilon$-DP. However, it turns out that there is another problem in this case, which is that $f(x)$ has a pole at $x = 0$. We use the fact that for $0 < x \ll \sqrt{|\nu| + 1}$, as $x \to 0^+$, $K_\nu(x)$ is asymptotically similar to

$$K_\nu(x) \sim \begin{cases} -\log(x) & \text{if } \nu = 0 \\ \frac{\Gamma(\nu)}{2}(2/x)^{|\nu|} & \text{if } \nu \neq 0, \end{cases}$$

where $\gamma$ is a constant (Abramowitz and Stegun, 1965, Chapter 9). Then

$$\lim_{y \to 0^+} f(y) = \left(\frac{y}{2}\right)^{\nu/2} K_\nu(\sqrt{2y}) \propto \lim_{y \to 0^+} \begin{cases} \exp(-\sqrt{y}) & \text{if } d = 1 \\ -\frac{1}{2}\log(2y) & \text{if } d = 2 \\ (y/2)^{\nu/2}(\frac{2}{\sqrt{2y}})^{|\nu|} & \text{if } d \geq 3 \end{cases}$$

From this, we see that the limit is finite when $d = 1$, but infinite when $d \geq 2$. So, the multivariate Laplace distribution cannot be used to achieve $\epsilon$-DP for $d \geq 2$.

While we may have supposed that the multivariate Laplace distribution would be well suited for $\epsilon$-DP, in fact it seems that the $K$-norm mechanism, introduced by Hardt and Talwar (2010), is a better generalization of the Laplace mechanism, since it is carefully tuned for privacy.

**Example 4** ($K$-Norm Mechanism). For any norm $\|\cdot\|_K$, the $K$-norm mechanism with mean $\mu$ draws from the density proportional to $\exp(-\|x - \mu\|_K)$. For norms of the form $\|x\| = \sqrt{x^\top \Sigma^{-1} x}$, the $K$-norm mechanism is an elliptical distribution, with density is proportional to $f((x-\mu)^\top \Sigma^{-1}(x-\mu))$, where $f(y) = \exp(-\sqrt{y})$. First note that there is no concern about poles, since $f(0)$ is finite.

For any $c \geq \Delta$, we have that $\dfrac{\exp(-\sqrt{(c-\Delta)^2})}{\exp(-\sqrt{c^2})} = \dfrac{\exp(-(c-\Delta))}{\exp(-c)} = \exp(\Delta)$, which is constant. This suggests that this distribution is especially suited for $\epsilon$-DP.

It is well known in the DP community that Gaussian noise cannot be used to achieve $\epsilon$-DP. We show in the next example how Theorem 3 can be used to easily verify this fact.

**Example 5** (Multivariate Normal). The density of a multivariate normal $N(\mu, \Sigma)$ has density proportional to $f((x - \mu)^\top \Sigma^{-1}(x - \mu))$, where $f(y) = \exp(-y/2)$. If $\Delta > 0$, then

$$\lim_{c \to \infty} \exp\left(-(c-\Delta)^2/2\right) / \exp\left(-c^2/2\right) = \lim_{c \to \infty} \exp\left(\frac{1}{2}\left(c^2 - [c^2 - 2c\Delta + \Delta^2]\right)\right) = \infty.$$

The previous result confirms that the tails of the Normal distribution are too light to achieve $\epsilon$-DP. In contrast with the previous example, we show next that the multivariate $t$-distribution can achieve $\epsilon$-DP, but its tails are maybe "over-kill".

**Example 6** (Multivariate $t$-distribution). A $d$ dimensional $t$ random vector with degrees of freedom $\nu > 1$, denoted $t_\nu^d(\mu, \Sigma)$ has density proportional to $f((x - \mu)^\top \Sigma^{-1}(x - \mu))$, where

$$f(y) = [1 + y/\nu]^{-(\nu+d)/2}.$$

We check the limit: $\lim_{c \to \infty} \dfrac{[1 + (c-\Delta)^2/\nu]^{-(\nu+d)/2}}{[1 + c^2/\nu]^{-(\nu+d)/2}} = \lim_{c \to \infty} \left[\dfrac{1 + c^2/\nu}{1 + (c-\Delta)^2/\nu}\right]^{(\nu+d)/2} = 1.$

Since the limit is finite, we know that there is a finite supremum. We solve $\dfrac{d}{dc}\left[\dfrac{1 + c^2/\nu}{1 + (c-\Delta)^2/\nu}\right] = 0$, and find that the unique solution in $[\Delta, \infty)$ is $c = \frac{1}{2}\left(\Delta + \left(\sqrt{\Delta^2 + 4\nu}\right)\right)$. Plugging this into $\left[\dfrac{1+c^2/\nu}{1+(c-\Delta)^2/\nu}\right]^{(\nu+d)/2}$ gives us the value of $\exp(\epsilon)$.

We end this subsection with a result for the original infinite dimensional problem: if $\mathbb{X}$ is infinite dimensional, then Theorem 3 can be used to achieve $\epsilon$-DP for a set of $d$ linear functionals from $\mathcal{K}$.

**Corollary 1.** *Assume $\mathbb{X}$ is an LCS of potentially infinite dimension. Let $T : \mathcal{D} \to \mathbb{X}$ be a summary with finite sensitivity $\Delta < \infty$ with respect to an elliptical noise $X \in \mathbb{X}$. Then for any distinct $g_i \in \mathcal{K}$ for $i = 1, \ldots, d$, the density of $\{g_i(\widetilde{T}_D)\}$ is proportional to $f(\sigma^{-2}(x - \mu_D)\Sigma^{-1}(x - \mu_D))$, where $\mu_D = \{g_i(T_D)\}$, $\Sigma = \{C(g_i, g_j)\}$, and $f : [0, \infty) \to [0, \infty]$ is a monotonically decreasing function depending on $d$ and the elliptical family, but not the specific $g_i$. If $f(0) < \infty$, and property (2) of Theorem 3 holds, then $\{g_i(\widetilde{T}_D)\}$ satisfies $\epsilon$-DP, where $\exp(\epsilon) = \sup\limits_{c \geq \sigma^{-1}\Delta} \dfrac{f((c - \sigma^{-2}\Delta)^2)}{f(c^2)} < \infty.$*

The key point of Corollary 1 is that there is a universal $\sigma$ such that $\widetilde{T}_D$ achieves $\epsilon$-DP when evaluated on any $d$ linear functionals. Unfortunately, it does depend on $d$, and as we will see in the next section, there is no finite $\sigma$ that can guarantee $\epsilon$-DP for arbitrary $d$ when using an elliptical perturbation.

## 4.2 Impossibility in Infinite Dimensional Spaces

In the previous subsection we gave a condition to check whether an elliptical distribution can be used to satisfy $\epsilon$-DP in finite dimensional spaces. It is natural to suppose that a similar property holds in infinite dimensional spaces. However, our main result in this section is that no elliptical distribution satisfies $\epsilon$-DP in infinite dimensional spaces. The intuition behind this result is that by

Corollary 1, any elliptical process can be expressed as a random mixture of Gaussian processes, but in infinite dimensional spaces, the mixing variable $V$ is actually measurable with respect to the infinite dimensional process. That is, if one observes $\tilde{T}_D = T_D + \sigma X$, then with probability one, the mixing random variable $V$ can be computed from $\tilde{T}_D$. This is because one can pool small amounts of information across an infinite number of dimensions estimate $V$ (even though $X$ still isn't observable). So, the noise from any elliptical distribution is equivalent (as far as privacy goes) to adding noise from a Gaussian process, which Mirshani et al. (2017) show only satisfies $(\epsilon, \delta)$-DP, a weaker notion of differential privacy than $\epsilon$-DP.

**Theorem 4.** *Consider a summary $T : \mathcal{D} \to \mathbb{X}$ and let $\tilde{T}_D = T_D + \sigma X$, where $X$ is a centered elliptical distribution and $T_D := T(D)$. If $\mathbb{X}$ is infinite dimensional, the image $T(\mathcal{D})$ is a not a singleton, and $C$ does not have finite rank, then $\tilde{T}_D$ will not achieve $\epsilon$-DP for any choice of $\sigma$.*

*Proof Sketch.* Consider functionals $g_i \in \mathcal{K}$ such that $C(g_i, g_j) = \delta_{ij}$. The estimator $V_n = \frac{1}{n} \sum_{i=1}^{n} g_i(\tilde{T}_D)^2$ converges to $V^2$ with probability 1 as $n \to \infty$, recovering $V$ from $\tilde{T}_D$. $\qquad\square$

Fortunately, elliptical distributions can still achieve $(\epsilon, \delta)$-DP. However, we run into a bit of an odd philosophical issue since the mixing coefficient $V$ can be computed from $\tilde{f}(D)$. So, the mechanism can be viewed as drawing from a mixture of Gaussian processes, but after observing the output the user knows exactly from which Gaussian distribution the noise came from.

**Theorem 5.** *Let $X$ be a centered elliptical process over $\mathbb{X}$ and $T : \mathcal{D} \to \mathbb{X}$ has sensitivity $\Delta$. Then for any $\epsilon > 0$ and $\delta > 0$,*

$$\tilde{T}_D = T_D + \sigma X, \qquad with \qquad \sigma^2 \geq \frac{2 \log(2/\delta')}{\epsilon^2} \Delta^2$$

*achieves $(\epsilon, \delta)$-DP, where $\delta'$ satisfies $\delta = 2 M_V(\log(\delta'/2))$ and $M_V$ is the moment generating function of mixing coefficient $V$, as defined in Theorem 1.*

In Theorem 5, $\delta'$ represents the DP that would be achieved under the Gaussian mechanism, thus one will end up with better privacy if $\delta < \delta'$. In addition, for $\delta' \in (0, 1)$, $\log(\delta'/2) < 0$, so $M_V(\log(\delta'/2))$ is finite and all quantities are well defined. The proof of Theorem 5 is similar to the proof of Mirshani et al. (2017, Theorem 3.3), and is in the Supplementary Materials.

## 5 Discussion

In this work we considered a new class of additive privacy mechanisms based on elliptical distributions. We also presented a number of foundational results concerning the equivalence/orthogonality of elliptical distributions. These mechanisms were considered under the general assumption that the summary resides in a locally convex space, allowing for a wide range of applications from classic multivariate statistics to nonparametric statistics and functional data analysis. Surprisingly, we show that while many elliptical distributions may be used for pure DP in finite dimensions, none are capable of achieving it in infinite dimensions. This is due to the close connection between Gaussian processes and elliptical processes, and both can only achieve approximate DP in infinite dimensions.

This work also highlights the need for more tools when the statistical summaries are complex objects such as functions. Properties that hold in finite dimensions may not hold in infinite dimensions in some surprisingly subtle ways. Practically, this can mean that either one does not have the desired level of protection against privacy disclosures, or that one has to add enormous amounts of noise to achieve pure DP.

While this paper has focused on the question of whether an elliptical distribution satsfies DP, we do not address the utility of these mechanisms. There is already evidence that elliptical distributions are useful for different applications (i.e. Awan and Slavković, 2019; Bun and Steinke, 2019), but further work establishing utility guarantees for elliptical distributions is needed.

## Footnotes

*Research supported in part by NSF DMS 1712826, NSF SES 1853209, and the Simons Institute for the Theory of Computing at UC Berkeley.

†Research supported in part by NSF SES-153443 and NSF SES-1853209.

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
