[Supplementary Material]

# Elliptical Perturbations for Differential Privacy
# Supplementary Materials

**Matthew Reimherr**
Department of Statistics
Pennsylvania State University
University Park, PA 16802
mreimherr@psu.edu

**Jordan Awan**
Department of Statistics
Pennsylvania State University
University Park, PA 16802
awan@psu.edu

*Proof of Lemma 1.* Suppose that $P^1(A) = 0$. By definition of conditional measures we have that $P^1(A) = \int P^1_t(A) dQ^1(t)$. However, if the above is zero, then it must be the case that $P^1_t(A) = 0$ for all $t$ except possibly on a set of $Q^1$ measure zero. By the assumed equivalence of $Q^1$ and $Q^2$ as well as the conditional measures, it follows that $P^2_t(A) = 0$ for all $t$ except possibly on a set of $Q^2$ measure zero. Therefore $P^2(A) = \int P^2_t(A) dQ^2(t) = 0$. Since we can swap the roles of $P^1$ and $P^2$, it follows that they agree on the zero sets. □

*Proof of Theorem 2.* The first direction is easier. Namely, if $\mu_1 - \mu_2$ resides in the Cameron-Martin space of $C$ then it resides in the Cameron-Martin space of any $vC$ for $v > 0$. To see this, note that $C$ and $vC$ induce equivalent norms; the latter is just scaled by $v$ compared to the former. Since the norms are equivalent, the resulting Cameron-Martin spaces are the same. Thus, conditioned on the mixture $V = v$, the measures are equivalent for all $v$ (Bogachev, 1998, Theorem 2.4.5 ). Thus Lemma 1 implies they are equivalent.

The harder part is the reverse. If the mixture is discrete, i.e. $V$ takes on at most a countable number of values, then we could, in principle, piece together a countable number of appropriate spaces (since $\sigma$-algebras are closed under countable unions). However, when $V$ is continuous, this approach won't work as it requires considering an uncountable number of sets. Thus, we have to be more explicit in terms of our construction. We consider, without loss of generality, $X_1 \sim \mathcal{E}(0, C, \psi)$ versus $X_2 \sim \mathcal{E}(\mu, C, \psi)$ where $\mu$ is not in $\mathbb{H}$, the Cameron-Martin space of $C$. To show that the two measures are orthogonal, it is enough to show that, for any fixed $\epsilon \in (0, 1)$ we can construct a set $A$ such that $P(X_1 \in A) \geq 1 - \epsilon$ while $P(X_2 \in A) \leq \epsilon$. Since $\mu$ is not in $\mathbb{H}$, it implies that the functional $T_\mu : \mathcal{K} \to \mathbb{R}$ defined as $T_\mu(f) = f(\mu)$ is not continuous, or equivalently, it is not bounded. So we can construct a sequence $g_1, g_2, \cdots \in \mathcal{K}$ such that $T_\mu(g_i) = g_i(\mu) \to \infty$ as $i \to \infty$, but $\|g_i\|_{\mathcal{K}} = 1$.

Now the random variable $g_i(X_1) \in \mathbb{R}$ has the same distribution as $V g_i(Z)$ and $g_i(X_2)$ the same as $g_i(\mu_i) + V g_i(Z)$, where $g_i(Z)$ has, by construction, a standard normal distribution. Let $c_\epsilon$ be a finite constant such that
$$P(V g_i(Z) \leq c_\epsilon) \geq 1 - \epsilon,$$
which does not depend on $i$ since $g_i(Z)$ is standard normal for all $i$. Define $A_i$ as
$$A_i = \{x \in \mathbb{X} : g_i(x) \leq c_\epsilon\},$$
which is a measurable set since any element of $\mathcal{K}$ is either an element of $\mathbb{X}^*$ or an appropriate limit. Then, for any $c_\epsilon$ we have that
$$P(g_i(X_2) \in A_i) = P(g_i(\mu) + V g_i(Z) \leq c_\epsilon) \to 0 \qquad \text{as} \qquad i \to \infty,$$
since $g_i(\mu) \to \infty$. So, we can choose $i$ and $A = A_i$ such that $P(g_i(X_2) \in A_i) \leq \epsilon$ and thus the distributions of $X_1$ and $X_2$ must be orthogonal. □

*Proof of Theorem 3.* Since $\sigma^2\Sigma$ is also a valid covariance matrix, we can always combine the two into one matrix, and thus, without loss of generality, we take $\sigma = 1$. To show $\epsilon$-DP it suffices to check that the ratio of the densities is bounded by $\exp(\epsilon)$, since the two are equivalent (Awan et al., 2019). Let $D, D' \in \mathcal{D}$ be fixed adjacent databases, $D \sim D'$. Denote $\mu_1 = T(D)$ and $\mu_2 = T(D')$ and then the ratio of densities is given by

$$\sup_{x \in \mathbb{R}^d} \frac{f((x - \mu_1)^\top \Sigma^{-1}(x - \mu_1))}{f((x - \mu_2)^\top \Sigma^{-1}(x - \mu_2))}.$$

Since $\Sigma$ has full rank, we can make the following change of variables without changing the maximization problem: $y = \Sigma^{-1/2}(x - \mu_2)$, which yields

$$\sup_{y \in \mathbb{R}^n} \frac{f((\Sigma^{1/2}y + \mu_2 - \mu_1)^\top \Sigma^{-1}(\Sigma^{1/2}y + \mu_2 - \mu_1))}{f(y^\top y)}.$$

Let $u = \Sigma^{-1/2}(\mu_2 - \mu_1)$, then we equivalently have the maximization

$$\sup_{y \in \mathbb{R}^n} \frac{f((y - u)^\top (y - u))}{f(y^\top y)}.$$

Next, notice that for any $y$, we can increase the ratio by rotating $y$ to point in the direction of $u$. This is because $f$ is decreasing and

$$(y - u)^\top (y - u) = y^\top y - 2y^\top u + u^\top u,$$

will be made smaller if $y^\top u$ is made larger, and the largest it can be while fixing the length $\|y\|_2 = c$ is when $y = cu/\|u\|_2$. So, we can express the maximization problem as

$$\sup_{c \geq 0} \frac{f(c^2 - 2c\|u\| + \|u\|^2)}{f(c^2)} = \sup_{c \geq 0} \frac{f((c - \|u\|)^2)}{f(c^2)}.$$

Since $f$ is monotonically decreasing the above is finite if and only if $f(0) < \infty$ and $\limsup_{c \to \infty} f((c - \|u\|)^2)f(c^2)^{-1}$ is finite. We can also restrict the supremum to $c \geq \|u\|$ as the supremum will never occur when $0 \leq c < \|u\|$. To see this, consider $a \in \mathbb{R}$ such that $0 \leq a < \|u\|$, and its reflection about $\|u\|$ given by $b = 2\|u\| - a$, which satisfies $b > \|u\|$. Then we have the numerators are the same, $f((a - \|u\|)^2) = f((b - \|u\|)^2)$, but the denominators satisfy $f(b^2) \leq f(a^2)$ since $f$ is strictly decreasing, which implies the ratio at $c = b$ is larger than at $c = a$.

Finally, $u$ still depends on $\mu_1$ and $\mu_2$. However, $\|u\| \leq \Delta$ and using that $f$ is monotonially decreasing, we have

$$\sup_{c \geq \|u\|} \frac{f((c - \|u\|)^2)}{f(c^2)} \leq \sup_{c \geq \Delta} \frac{f((c - \Delta)^2)}{f(c^2)} := \exp\{\epsilon\}.$$

To obtain different values of $\epsilon$ we can replace $\Delta$ with $\sigma^{-1}\Delta$ and adjust $\sigma$ until the desired $\epsilon$ is achieved. $\qquad\square$

*Proof of Theorem 4.* We can assume that $T_D \in \mathbb{H}$ as otherwise $\widetilde{T}_D$ and $\widetilde{T}_{D'}$ are orthogonal (in which case it is trivial that DP doesn't hold). The key issue is that, in infinite dimensions, one learns too much about the mixing coefficient. In particular, consider functionals $g_i \in \mathcal{K}$ such that $C(g_i, g_j) = \delta_{ij}$. One can find an infinite number of such functionals as long as $C$ does not have finite rank. Then consider

$$V_n = \frac{1}{n}\sum_{i=1}^{n} g_i(\widetilde{T}_D)^2 = \frac{1}{n}\sum_{i=1}^{n} g_i(T_D)^2 + 2V\frac{1}{n}\sum_{i=1}^{n} g_i(Z) + V^2\frac{1}{n}\sum_{i=1}^{n} T_i(Z)^2.$$

Now notice that, by Parceval's identity, $\sum_{i=1}^{n} g_i(f(D))^2 \leq \|f(D)\|_{\mathcal{K}}$ and that $g_i(Z)$ are iid standard normal. Thus the first two terms converge to 0 with probability 1, while the second term converges to $V^2$. So, if we observe $\widetilde{T}_D$ then we can reconstruct $V$ perfectly (since $V > 0$) and thus, in the DP calculation it can be treated as fixed, $V = v$. Now notice that $\widetilde{T}_D | V = v$ is simply Gaussian and does not achieve $\epsilon$-DP, meaning, for any $\epsilon > 0$ we can find a set $A_v$ where $P(\tilde{f}(D) \in A_v | V = v) > e^\epsilon P(\tilde{f}(D') \in A_v | V = v)$. Thus the mechanism is not $\epsilon$-DP. $\qquad\square$

*Proof of Theorem 5.* Notice that conditioned on $V = v$, we have that $\tilde{f}(D) = f(D) + \sigma v Z$ and $\sigma^2 = \dfrac{2\log(2/\delta')}{\epsilon^2}\Delta^2$. So, the noise is scaled by $v$. If we absorb this into the $\delta'$ then $v\log(2/\delta') = \log((2/\delta')^v)$. Finally,

$$P(\tilde{f}(D) \in A) = \int P(\tilde{f}(D) \in A | V = v) d\psi(v) \leq e^\epsilon P(\tilde{f}(D') \in A) + 2\,\mathrm{E}[(\delta'/2)^V]$$

$$= e^\epsilon P(\tilde{f}(D') \in A) + 2M_V(\log(\delta'/2)) = e^\epsilon P(\tilde{f}(D') \in A) + \delta. \qquad \square$$