[Reviews · NeurIPS 2019]

Reviewer 1



The paper is very well written and easy to follow. The question the authors are trying to solve is interesting: are there mechanisms that can achieve (eps, 0) differential privacy in general linear spaces. While the answer provided by the authors is still not conclusive they show that a very general class of mechanisms (which achieve (eps, 0) differential privacy in finite dimensions) is in fact only (eps, delta) DP in infinite dimensions. The ideas and proofs in the paper are very insightful. I have two comments that I believe need to be addressed before approving the paper for publication: 1. I believe Lemma 1 assumes the existence of regular conditional probability measure. Otherwise, the decomposition of P^1 as a an integral wouldn't hold. However the existence of a regular conditional measure is not guaranteed specially in such a high dimensional setting. Can the authors show that the conditional measure exists for the proposed process? 2. In the proof of Theorem 2. The authors say that because v (mu_1 - mu_2) is in the Cameron-Martin space for all v then it follows that the conditional distributions must be equivalent. I assume this is the case because v mu_1 and v_mu2 are gaussian measures and according to the authors two Gaussian measures are either equivalent or orthogonal. However, the authors fail to provide the conditions for them to be equivalent. Therefore, the proof doesn't quite follow.

Reviewer 2



The paper proposes to use elliptical distributions for adding perturbations to achieve differential privacy (DP) in infinite dimensional and multivariate settings. This is a significant and original contribution; an understanding of how to achieve DP in these settings has been lacking. The paper is has important theoretical contributions, but is entirely theoretical. There is no motivation of a real use case provided. In what practical statistical or machine learning problems does the need for this arise? While the paper is fairly readable, it is really directed to a reader well-versed in measure theory. Many parts of the paper could improve with explanations or intuitive descriptions approachable by a more general reader. Understanding why current methods do not achieve DP is important, and the authors do address this. However this is not presented in an easily understandable manner. More clarity is needed in the section on the impossibility result. The jump from Theoreom 4 to Theorem 5 is not clear -one is an impossibility result, the other gives a result on achieving DP. Further, the paper abruptly ends in the middle of a section.

Reviewer 3



The paper studies random perturbation mechanisms for private query release in abstract topological vector spaces. The notion of elliptical perturbations, defined by the mean, a positive definite kernel and a scaling function, is studied with several results in both finite and infinite-dimensional spaces. The basic rationale behind elliptical perturbations is that: for infinite-dimensional spaces, add isotropic noise will usually result in a point outside the space. So elliptical perturbations are introduced. This is not the first time such idea being explored: Gaussian process noises has been utilized in Hilbert spaces. However, it is still worthwhile to characterize the possibilities and impossibilities for infinite-dimensional spaces in a precise way, which is the main contribution of the paper. On the other hand, the author did not show how the abstract results can benefit any particular examples. In differential privacy research, it is important to compare the utility of the mechanism, instead of only analyzing the privacy guarantees. The author did not show why the mechanisms constructed from the abstract conditions could lead to possibly more accurate private answers. Minor comments: The remark following Theorem 4 at page 8 is based on incorrect understanding of differential privacy. The statement "One is achieving a random level of privacy which average out to the desired level. In fact, the actual level of DP can be determined once the output is released" makes no sense. DP is a property of a randomized algorithm which takes into account all the randomization. An algorithm either satisfies DP or not.

[Author Response · NeurIPS 2019]

Reviewer #1 asks a technical question regarding the existence of a regular conditional measure in Lemma 1. The reviewer is correct that we are implicitly assuming the conditional densities exist, which need not be the case in more complex spaces. However, we are only using Lemma 1 regarding elliptical processes (Thm 2). For elliptical processes the existence is not an issue, since the conditional distribution can be written explicitly based on Thm 1, which implies that the conditional distribution is a gaussian process. We will clarify in Lemma 1 that this condition is required, and explain why the conditional distributions exist for elliptical processes in the final manuscript.

While not necessary for the study of elliptical processes, with a few more assumptions on the space, we can ensure that an arbitrary probability measure has a regular conditional measure. Recall that conditional distributions exist as long as the measures are inner regular (Hoffman-Jorgensen, 1972). Since every probability measure over a complete separable metric space (equipped with the Borel sigma algebra) is Radon (e.g. THM A.3.11, Bogachev, 1998), and thus inner regular, restricting to such spaces removes this technical concern and still covers most applications of interest.

Reviewer #1 also asks how to determine whether two Gaussian measures are equivalent or orthogonal, a task that contributes to the proof of Theorem 2. We appreciate the reviewer raising this point. The claim follows from Theorem 2.4.5 of Bogachev (1998), which characterizes the equivalence/orthogonality of Gaussian measures with equal covariances. We will add this to the proof sketch and main proof in the manuscript and make it clear how the Theorem applies in our case. Briefly, (1) the covariance $vC$ leads to the same Cameron-Martin space for any nonzero $v$ since they induce equivalent norms (the $v$ just scales each norm) (2) two Gaussian measures with the same covariance $C$ are equivalent if the difference of their means lies in the Cameron-Martin space of $C$.

Reviewer #3 ask for more intuitive explanations throughout to better communicate our results to a reader with less measure theory background. We agree with this sentiment and will add more discussion and examples when technical concepts are introduced. Concepts that will be expounded upon include elliptical distributions (ln 30), function spaces (ln 37), equivalence of measures (ln 59), absolutely continuous (ln 104), Haussdorf (ln 122), Cameron-Martin theory (ln 140). We will also include discussion regarding the intuition and consequences of each of our technical results.

Both Reviewers #3 and #4 ask for concrete examples to motivate the use of elliptical distributions. We will include additional motivation to the introduction. Briefly, our motivation stemmed from the popularity of the Laplace mechanism over the Gaussian for achieving DP in $\mathbb{R}$, both of which are examples of elliptical distributions, as are certain instances of the $K$-norm mechanism. However, in dimensions greater than 1, neither the Gaussian nor Laplace provide $\epsilon$-privacy. A recent technical report by Bun & Steinke: "Average-Case Averages: Private Algorithms for Smooth Sensitivity and Mean Estimation", they explore several univariate distributions such as Cauchy, Student's T, Laplace Log-Normal, Uniform Log-Normal, Arsinh-Normal. We hope that our paper may help motivate the study of elliptical versions of these distributions. In Bun & Steinke, they found that Laplace Log-Normal distribution and Student's T distribution had the lowest variance for their problem.

Reviewer #4 points out that we focus on the question of whether an elliptical distribution satisfies DP, and do not address the utility of the mechanisms. It has already been demonstrated that different elliptical distributions are useful for different applications (i.e. Awan & Slavković (2019), Bun & Steinke (above)). The focus of this paper is to better understand when different elliptical distributions can be applied and what level of privacy they achieve in general vector spaces.

Reviewer #3 requests more exposition in Section 4.2, to explain the relation between Thm 4 and Thm 5. The intuition is as follows: In Thm 4, we show that the mixing random variable $V$ can be estimated arbitrarily well based on the output $\tilde{T}(D)$. This implies that the result of this mechanism is essentially the same as adding a gaussian process to $T$, once the output is observed. Since gaussian processes do not satisfy $\epsilon$-DP, neither can any elliptical process. However, gaussian processes do satisfy $(\epsilon, \delta)$-DP. Thm 5 explains what values of $\epsilon$ and $\delta$ a given elliptical process achieves, which is based on the same proof (Mirshani et al. 2017) used to prove that gaussian processes satisfy $(\epsilon, \delta)$-DP. We will include additional exposition in the final version to explain these points.

Related to the previous point, reviewer #4 notes out that the remark following Thm 4 is misleading. We meant to explain that once the sample is released, it is essentially a sample from a gaussian process. However, the covariance is scaled by the variable $V$, which is known after release.

Reviewer #3 remarks that the paper ends after Section 4.2, which is stylistically undesirable. As the final version allows an additional page, we will include a section with discussion/future directions.



[Meta-Review · NeurIPS 2019]

This is a purely theoretical paper that includes results about when it is and is not possible to achieve differential privacy in infinite-dimensional spaces. The reviewer scores were 6, 7, 8; one reviewer increased their score during the discussion phase. The reviewers found the theoretical contributions insightful, significant, and original. Some critiques were raised, including accessibility to an audience not well versed in measure theory, and lack of concrete examples or applications. The author response answered a number of specific questions and promised to provide more exposition and concrete examples to address some of the reviewer concerns, which the AC encourages them to do. Overall, the consensus is that this is a good paper.